# Advancing science or advancing careers? Researchers' opinions on success indicators

**Noémie Aubert Bonn**[ID]\*, **Wim Pinxten**[ID]

Healthcare and Ethics Research Group, Faculty of Medicine and Life Sciences, Hasselt University, Hasselt, Belgium

\* noemie.aubertbonn@uhasselt.be

**Data Availability Statement:** All relevant data are within the manuscript and its Supporting Information files.

**Funding:** This work was supported by internal funding from Hasselt University Bijzonder

## Abstract

The way in which we assess researchers has been under the radar in the past few years. Critics argue that current research assessments focus on productivity and that they increase unhealthy pressures on scientists. Yet, the precise ways in which assessments should change is still open for debate. We circulated a survey with Flemish researchers to understand how they work, and how they would rate the relevance of specific indicators used in research assessments. We found that most researchers worked far beyond their expected working schedule. We also found that, although they spent most of their time doing research, respondents wished they could dedicate more time to it and spend less time writing grants and performing other activities such as administrative duties and meetings. When looking at success indicators, we found that indicators related to openness, transparency, quality, and innovation were perceived as highly important in advancing science, but as relatively overlooked in career advancement. Conversely, indicators which denoted of prestige and competition were generally rated as important to career advancement, but irrelevant or even detrimental in advancing science. Open comments from respondents further revealed that, although indicators which indicate openness, transparency, and quality (e.g., publishing open access, publishing negative findings, sharing data, etc.) should ultimately be valued more in research assessments, the resources and support currently in place were insufficient to allow researchers to endorse such practices. In other words, current research assessments are inadequate and ignore practices which are essential in contributing to the advancement of science. Yet, before we change the way in which researchers are being assessed, supporting infrastructures must be put in place to ensure that researchers are able to commit to the activities that may benefit the advancement of science.

## Introduction

The way we define and evaluate scientific success impacts the way in which research is performed [1]. Yet, definitions of success in science are ambiguous and have raised many debates in the past few years. The San Francisco Declaration on Research Assessments [2], the Leiden Manifesto [3], or the Metric Tide [4] are all examples that denounced the inadequacy of the

Onderzoeksfonds (BOF), grant number 15NI05.The funders had no role in the study design, data collection and analysis, decision to publish, or preparation of the manuscript.

**Competing interests:** The authors have declared that no competing interests exist.

metrics currently used for research assessments. Most critics argue that current metrics are reductionistic and inappropriate for individual evaluations. But despite increasing criticism, alternative assessments are difficult to find [e.g., 5]. Some argue that narratives and subjectivity must be reintroduced in research assessments, while other support the need for new metrics to broaden the scope of evaluations. As the approaches that need to be taken are still disputed, a few institutions and funding agencies have taken the lead in exploring new ways to evaluate researchers [see for e.g., 6–8].

Research assessments do not only depend on institutions and funding agencies. When assessments are organised by institutions and agencies, researchers are often the ones who act as referee. Consequently, changing research assessments does not only require new regulations, guidance, and policies from institutions, but also cultural changes from the research floor.

Last year, we explored definitions of success in biomedical research in Flanders, Belgium [9, 10]. Using interviews and focus groups, we captured the perspectives of different research actors on what determines success in science. Actors included research institutions, research funders, scientific editors or publishers, researchers, research students, and several other actors who play an important role in academic research. Although interviewees largely agreed on what currently constitutes success in science, their opinions also conflicted in some aspects. Oftentimes, the same indicator of success would raise opposite reactions, with some proposing that one indicator was paramount to advancing science and others arguing that the same indicator threatened research integrity. Using the indicators of success which were mentioned in the interviews and focus groups, we built a brief survey to better understand the importance of each indicator in advancing science, in advancing one's career, and in yielding personal satisfaction.

## Materials and methods

### Tool

Using Qualtrics WM, we built the survey from themes extracted in past interviews and focus groups [9]. More specifically, we built the survey questions on the indicators which were thought to play a role in acquiring success in science, but which also raised conflicting opinions. The resulting survey assesses the impact of each indicator on (i) career advancement, (ii) scientific advancement, and (iii) personal satisfaction. We refined the survey questions and statements several times by consulting a few experts and colleagues as well as researchers with experience in similar questionnaires. We tested the survey with close colleagues (4 PhD students, and 4 senior researchers) to ensure clarity and relevance of the questions. The final version of the survey includes 18 statements whose impact is assessed on each of the three pillars. Table 1 showcases the final statements which figured in the survey alongside the available answer options.

Beyond the 18 statements included in the survey, we asked a few demographic questions, such as gender, university and faculty of affiliation, current position and seniority, and number of publication. We also included a series of questions on time management which asked the average number of hours worked each week; the percentage of time dedicated to 'research', 'teaching', and 'other tasks'; and in greater detail, the percentage of time dedicated to 'direct student supervision', 'hands on research work', 'staying up to date', 'writing papers', 'reviewing', 'grants writing', and 'other tasks'. The full printout of the survey is available in S1 File.

### Recruitment

To enable broad recruitment in the Flemish academic research landscape, we initially considered sharing the survey using Web of Science corresponding emails from Flemish authors available in the database. Nevertheless, after consulting the Data Protection Officer at our institution, we found that this recruitment method might not be fully compliant with the new

**Table 1. Statements included in the survey and answer options.**

| | | |
|---|---|---|
| Publishing papers is . . . | □ essential | . . .in advancing my career |
| Publishing in high impact journals is . . . | □ important | |
| Publishing commentaries or editorials is . . . | □ irrelevant | |
| Publishing more papers than others is . . . | □ unfavorable | |
| Publishing open access is . . . | □ detrimental | |
| Peer reviewing is . . . | | |
| Replicating past research is . . . | | |
| Publishing findings that did not work (i.e., negative findings) is . . . | | |
| Sharing your full data and detailed methods is . . . | □ essential | . . .in advancing science |
| Reviewing raw data from students and collaborators is . . . | □ important | |
| Conducting innovative research with a high risks of failure is . . . | □ irrelevant | |
| Connecting with renowned researchers is . . . | □ unfavorable | |
| Collaborating across borders, disciplines, and sectors is . . . | □ detrimental | |
| Getting cited in scientific literature is . . . | □ essential | . . .to my personal satisfaction |
| Having your papers read and downloaded is . . . | □ important | |
| Having public outreach (e.g., social media, news, etc.) is . . . | □ irrelevant | |
| Having your results used or implemented in practice is . . . | □ unfavorable | |
| Having luck is . . . | □ detrimental | |

European General Data Protection Regulation (GDPR; i.e., the purpose for sharing one's email address when publishing as first author does not imply an agreement to receive invitations to research surveys). Consequently, in respect of GDPR, we downsized our prospective sample and directly contacted the faculties of medicine and life science or equivalent to ask them to circulate our survey within their faculty.

We contacted Deans and Directors of doctoral schools from all five Flemish universities, namely Universiteit Antwerpen (UAntwerpen), Universiteit Gent (UGent), Universiteit Hasselt (UHasselt), Katholieke Universiteit Leuven (KU Leuven), and Vrije Universiteit Brussels (VUB). We further reached out to the Institute of Tropical Medicine Antwerp (ITM) and to the Interuniversity Microelectronics Centre (IMEC) using contacts we knew from within the institutes. Two universities distributed the survey with the entire faculty via email (UHasselt and UAntwerpen) and ITM agreed to distribute the survey internally to its researchers. One university preferred not to share the survey within its institution (KU Leuven) and our emails to one university remained unanswered (VUB). One university agreed to distribute the survey, be it not by distribution via a mailing list but by social media invitation to our survey (UGent). Given the latter, we composed an invitation which was shared by UHasselt's social media accounts (Twitter and Linked'in), and later shared by UGent's social media accounts. By sharing the survey publicly, we allowed anyone interested to participate, whether they were affiliated with a Flemish institution or not. We encouraged any re-tweets, likes, and shares to promote participation via snowballing strategies. Select contacts at IMEC and UGent also shared the survey within colleague groups. Finally, we further transferred the survey to select research groups and mailing lists with whom we are acquainted in Flanders. As a result, our participant group is diverse and spread out but also uneven, with numerous responses from institutions who shared the survey internally and few responses from institutions who relied on social media or snowballing. The survey was open from the 8th to the 31st of October 2019.

The project was approved by the committee for medical ethics (Comité voor Medische Ethiek) of the Faculty of Medical and Life Sciences of Hasselt University, protocol number cME2019/O3s. At the beginning of the survey, participants were referred to an information

sheet detailing how the data from the current project would be used (a printout of this information sheet is available at https://osf.io/78mgs/). By agreeing to participate, participants agreed that the data, which contains no identifiable information, would be published alongside the results of the study. Before sharing the data, we further removed specific information about the affiliation of researchers to ensure greater confidentiality. In our assessment, the study and curated data thus pose no risk of identification and privacy.

## Data analysis

For time distribution analyses, we used paired t tests in RStudio version 1.1.453 to compare the time respondents declared spending on different research activities to the time they wished they would spend on each of these activities.

For the dimensions of success indicators, we captured the views of participants using Likert scales with five options to rate the importance that each success indicator had on 'advancing science', 'advancing careers', and 'contributing to personal satisfaction' (Table 1). Before analysing our data statistically, we transformed the answer options to numerical values (i.e., detrimental = 1, unfavorable = 2, irrelevant = 3, important = 4, essential = 5). Nevertheless, the possibility of interpreting Likert Scale data as continuous data raises controversies, and our findings should be interpreted with caution. Indeed, since Likert scales provide no guarantee that the distance between each category is equal, many propose that they should be treated as ordinal data [11–13]. We decided to use repeated measures ANOVAs with Bonferroni post hoc test in SPSS Statistics version 25.0.0.0 to compare the average ratings of each statement's importance in i) advancing one's career, ii) advancing science, and iii) advancing one's personal satisfaction, but to also provide thorough visual depictions of our data (made using Tableau Desktop 2018.1 and Excel) and access to full data files to allow re-analysis and re-interpretation of our findings. Our survey data are available as in S3 File, and the codes used to analyse the data are available in S2 File.

## Results

### Data availability

The datasets are available in S3 File. To ensure confidentiality and to avoid inter-university comparisons (which we believe would provide no useful information at this point), we extracted the detailed affiliation from the dataset, leaving only the information stating whether the respondent was affiliated with a Flemish institution or not.

### Participants

In total, 126 participants completed the survey, two-thirds of which were either PhD students or post-doctoral / non-tenure-track researchers (Table 2). The gender distribution was well balanced (64 females, 60 males, and 2 prefer not to disclose). Almost 90% of participants were affiliated with Flemish institutions (n = 112), with the Universiteit Antwerpen, Universiteit Hasselt, and the Institute of Tropical Medicine Antwerp being the three most represented (Table 2). A few international participants also contributed to the survey. Most participants had below ten published papers and around three quarter of respondents had below 30 publications. Yet, the distribution of publication profiles was broad, and ranged until the maximum option available in the survey of 'over 210 published papers'.

### Time management

Almost three quarter of the respondents (n = 93) stated that they worked full time as a researcher or PhD student. On average, respondents who declared working full time worked

**Table 2. General demographics.**

| | | |
|---|---|---|
| **POSITION** | | |
| PhD Student | | 48 |
| PostDoc / Non-tenure-track researcher | | 36 |
| Tenure-track researcher / Professor | | 13 |
| Tenured researcher / Full professor | | 13 |
| Emeritus Professor | | 3 |
| Former researcher | | 6 |
| Other | | 7 |
| **GENDER** | | |
| Female | | 64 |
| Male | | 60 |
| Prefer not to say | | 2 |
| **AFFILIATION** | | |
| Affiliated with a Flemish institution | | 113 |
| | UAntwerpen | 31 |
| | UHasselt | 28 |
| | ITM | 18 |
| | KU Leuven | 11 |
| | UGent | 9 |
| | IMEC | 9 |
| | VUB | 4 |
| | Other | 3 |
| Affiliated with an institution outside Flanders | | 13 |
| **PUBLICATION PROFILE** | | |
| <10 peer-reviewed papers | | 58 |
| 10–30 peer-reviewed papers | | 40 |
| 30–60 peer-reviewed papers | | 9 |
| 60–90 peer-reviewed papers | | 4 |
| 90–120 peer-reviewed papers | | 3 |
| 120–150 peer-reviewed papers | | 6 |
| 150–180 peer-reviewed papers | | 0 |
| 180–210 peer-reviewed papers | | 3 |
| >210 peer-reviewed papers | | 3 |
| **TOTAL NUMBER OF PARTICIPANTS** | | 126 |

46.91 hours per week (median 46) but the distribution was very wide (Table 3). Among those who declared working full time as a researcher or a research student, 73 (78.5%) said that they worked more than 40 hours per week. Among these 73 respondents, 41 (44.1%) declared that they worked more than the European Union directive maximum of 48 hours per week [14], and 11 (11.8%) declared that they worked 60 hours per week or more until a maximum of 80 hours (fixed maximum in the survey). When including respondents who were not declared as full time researcher, the number of respondents declaring to work 60 or more hours per week rose to 18.

In the next questions, we targeted more specific research activities to understand how researchers distribute their research time. A first question asked respondents to distribute their time between three main pillars, namely 'teaching', 'research', and 'other'. A second question targeted more specific activities, namely 'direct student supervision', 'hands-on research work (e.g., lab work, data analysis)', 'staying up-to-date (e.g., reading, listening, building skills,

**Table 3. Reported weekly hours of work for respondents who declared working full time.**

| Position | N | Average time | Median | Min | Max |
|---|---|---|---|---|---|
| PhD student | 40 (39) | 46.30 (46.72) | 45 (45) | 30 (38) | 60 |
| Post-doctoral / Non-tenure track position | 27 | 46.59 | 46 | 40 | 70 |
| Tenure-track researcher / Professor | 9 (8) | 48.00 (54.00) | 50 (55) | 0 (42) | 65 |
| Tenured researcher / Full professor | 8 (7) | 51.13 (57.29) | 54.50 (59) | 8 (42) | 80 |
| Researcher in the past, but moved to another career | 4 | 48.75 | 49 | 46 | 51 |
| Other | 5 | 43.40 | 45 | 36 | 51 |
| | Total N 93 (90) | Overall average 46.91 (48.06) | Overall median 46 (46.5) | Overall min. 0 (36) | Overall max. 80 (80) |

Note: Numbers in parentheses exclude three answers which were not above 30h per week (i.e., 0, 8, and 30). We deemed that these answers may be outliers since respondents confirmed being employed full time as researchers, and thus the answers reflect a misunderstanding with our interpretation of full time research employment.

etc.)', 'writing papers', 'reviewing', 'grant writing', and 'anything else (e.g., administration, meetings, etc.)'. For both questions, we asked respondents to tell us the percentage of time they really spent on each activity, as well as the percentage of time they would like to attribute to each activity if they were in an ideal world.

The range of answers was very broad, as could be expected for this type of question. Since some categories raised different responses between respondents who worked full time as researchers and those who did not (i.e., 'research' and 'other' from the main pillars, and 'research work' and 'anything else' from the detailed categories), we decided to keep only full-time researchers (n = 93) for analyses of the questions on time distribution.

From the general pillars, we found that participants wished they could dedicate more time to 'teaching' and especially to 'research', but less time to 'other' activities. When looking at the detailed activities, we could see that participants generally hoped they could spend more time on 'hands-on research work', 'staying up to date', and 'writing papers'. On the other hand, respondents wished to spend less time on 'grant writing' and doing 'anything else (e.g. administration, meetings, etc.)'. The differences between real and ideal times for *'direct student supervision' and 'reviewing'* were not significant. Complete statistical results are available in Table 4. Fig 1 illustrates the distribution of answer for each activity.

## Impact of activities and indicators

In the last section of the survey, we asked participants to indicate the impact of 18 different research activities on i) advancing their career, ii) on advancing science, and iii) on their personal satisfaction (Table 1). Since our data comes from Likert scales we must be careful in interpreting our findings statistically (see more information about this in the methods section). Yet, we believed it worthy to maximally examine our data and conducted repeated measures ANOVAs while carefully complementing our analyses with thorough visual representations of the answers gathered so as to increase the comprehensibility of the data.

In Fig 2, we show the mean and median scores gathered for each of the 18 statements. We organized statements to allow a quick visual inspection of statements which were rated as highly important in advancing science, but less relevant in advancing one's career (North-East quadrant), and statements which were considered essential in advancing one's career, but of lesser importance in advancing science (South-Western quadrant). Except for the statement 'Having public outreach (e.g., social media, news, etc.)', all main effects were significant,

**Table 4. Comparison of reported percentage of time allocated of each task in reality to in an ideal world (how respondents wish it would be).**

| | Area | Sig. | Reality | Ideal world | Statistical results |
|---|---|---|---|---|---|
| General pillars | Teaching | * | Mean 11.9 | Mean 15.3 | Paired t(92) = 2.59 |
| | | | Median 10 | Median 15 | 95% CI 0.79, 5.96 |
| | | | | | p < 0.05 |
| | Research | *** | Mean 63.5 | Mean 72.6 | Paired t(92) = 4.48 |
| | | | Median 60 | Median 75 | 95% CI 5.10, 13.24 |
| | | | | | p < 0.001 |
| | Other | *** | Mean 24.6 | Mean 12.1 | Paired t(92) = −7.44 |
| | | | Median 20 | Median 10 | 95% CI −15.90, −9.20 |
| | | | | | p < 0.001 |
| Detailed activities | Direct student supervision | | Mean 11.5 | Mean 11.9 | Paired t(92) = 0.50 |
| | | | Median 10 | Median 10 | 95% CI −1.42, 2.37 |
| | | | | | p = 0.62 |
| | Hands on research work (e.g. lab work, data analysis, etc.) | *** | Mean 28.7 | Mean 36.1 | Paired t(92) = 5.15 |
| | | | Median 26 | Median 35 | 95% CI 4.55, 10.27 |
| | | | | | p < 0.001 |
| | Staying up to date (e.g. reading, listening, building skills, etc.) | ** | Mean 13.5 | Mean 16.8 | Paired t(92) = 2.81 |
| | | | Median 10 | Median 15 | 95% CI 0.96, 5.60 |
| | | | | | p < 0.01 |
| | Writing papers | ** | Mean 12.9 | Mean 15.8 | Paired t(92) = 3.07 |
| | | | Median 10 | Median 15 | 95% CI 1.01, 4.69 |
| | | | | | p < 0.01 |
| | Reviewing | | Mean 6.0 | Mean 6.2 | Paired t(92) = 0.30 |
| | | | Median 5 | Median 5 | 95% CI −1.03, 1.39 |
| | | | | | p = 0.77 |
| | Grant writing | *** | Mean 9.4 | Mean 6.0 | Paired t(92) = −3.70 |
| | | | Median 6 | Median 5 | 95% CI −5.22, -1.58 |
| | | | | | p < 0.001 |
| | Anything else (e.g. administration, meetings, etc.) | *** | Mean 18.0 | Mean 7.2 | Paired t(92) = −9.04 |
| | | | Median 15 | Median 5 | 95% CI −13.17, -8.42 |
| | | | | | p < 0.001 |

meaning that at least two of the dimensions (advancing science, advancing careers, or personal satisfaction) differed from one another. S1 Table shows all means and statistical results. Bonferroni post hoc test revealed that, in most cases, activities were rated differently on their impact in advancing science than on their impact in advancing one's career. Only one statement yielded similar scores in advancing one's career and in advancing science, namely 'Having your papers read and downloaded'. 'Publishing papers' was thought to be slightly more important in advancing one's career than in advancing science, but the difference was not so distinct (means of 4.52 and 4.37, respectively). Personal satisfaction, on the other hand, was at times closer to the impact on one's career, and at other times closer to the impact in advancing science. Specifically, 'Peer reviewing' and 'Collaborating across borders, disciplines, and sectors' were rated similarly on their importance to researchers' careers and personal satisfaction. While 'Peer reviewing' was low on both career and personal satisfaction, 'Collaborating across borders, disciplines, and sectors' was rated as highly important for both. 'Publishing in high impact journals', 'Publishing more papers than others', 'Connecting with renowned researchers', and 'Having luck' were rated as contributing more to advancing researchers' career than to either science and personal satisfaction. Finally, 'Having results used or implemented in

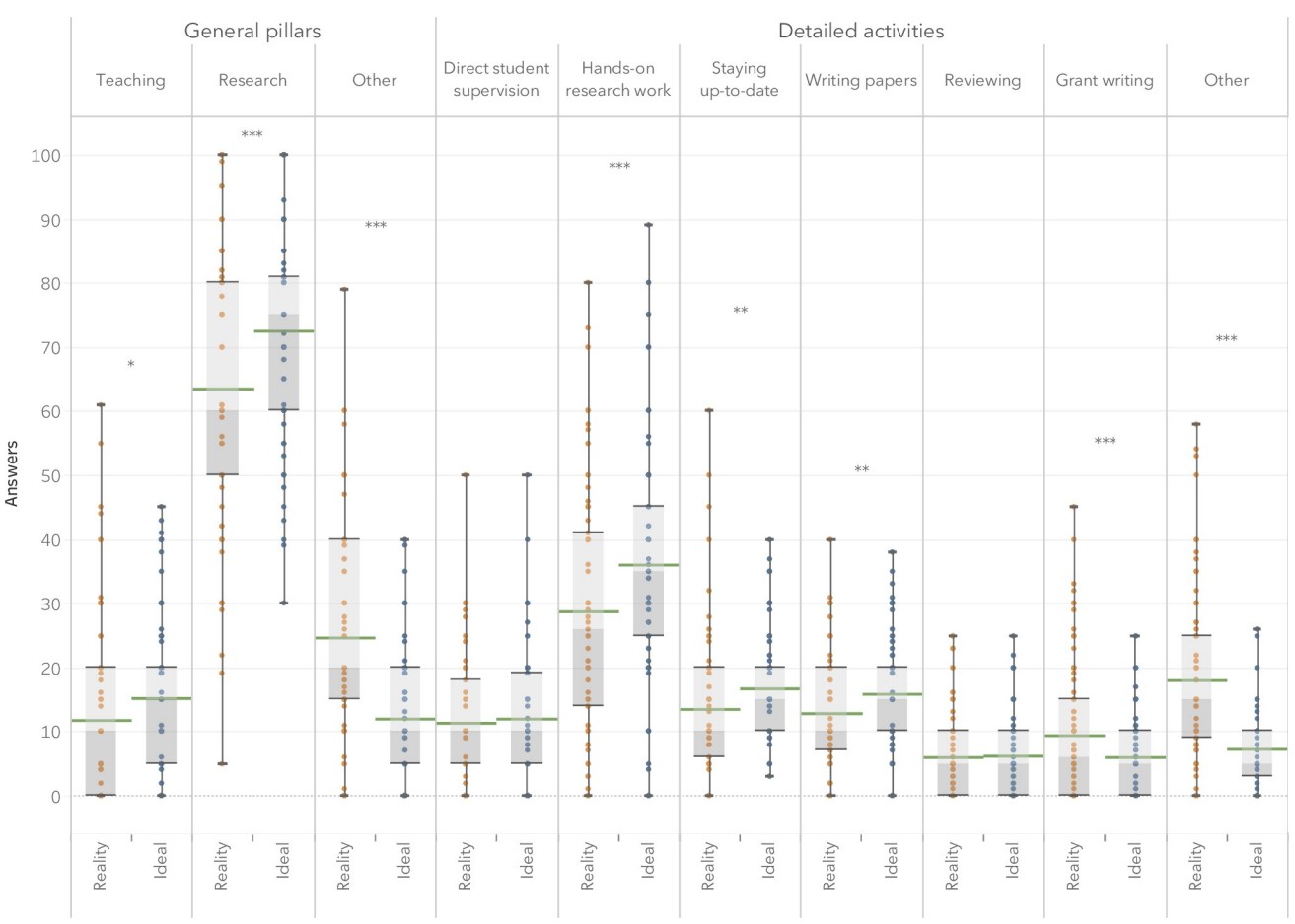

**Fig 1. Self-reported percentage of time spent on different research activities in reality and in an ideal world for respondents who declared working full time (n = 93).**

practice' was rated higher on both personal satisfaction and scientific advancement than on the impact it has on career advancement.

Taking our statements on a second level, a few trends become visible. First, it appears that various activities meant to promote openness and transparency (i.e., 'Publishing findings that did not work'; 'Sharing your full data and detailed methods'; 'Publishing open access'), quality assurance (i.e., 'Replicating past research'; 'Peer reviewing', and 'Reviewing raw data from students and collaborators'), and innovation (i.e., 'Conducting innovative research with a high risk of failure') were thought to be important to advancing science but significantly less important in advancing researchers' careers. Looking at the full range of answers in Fig 3, we can see that several respondents classified some of these activities as unfavorable or even detrimental in advancing their career. On the other hand, a few statements were rated as more important in advancing researcher's career than in advancing science. Among those, statements which relate to competition (i.e., 'Publishing more papers than others') and prestige (i.e., 'Getting cited in the literature', 'Publishing in high impact journals', and 'Connecting with renowned researchers') were most notable. As a general rule, respondents appeared to have most satisfaction from 'Collaborating across borders, disciplines, and sectors', and from 'Having their results implemented in practice'.

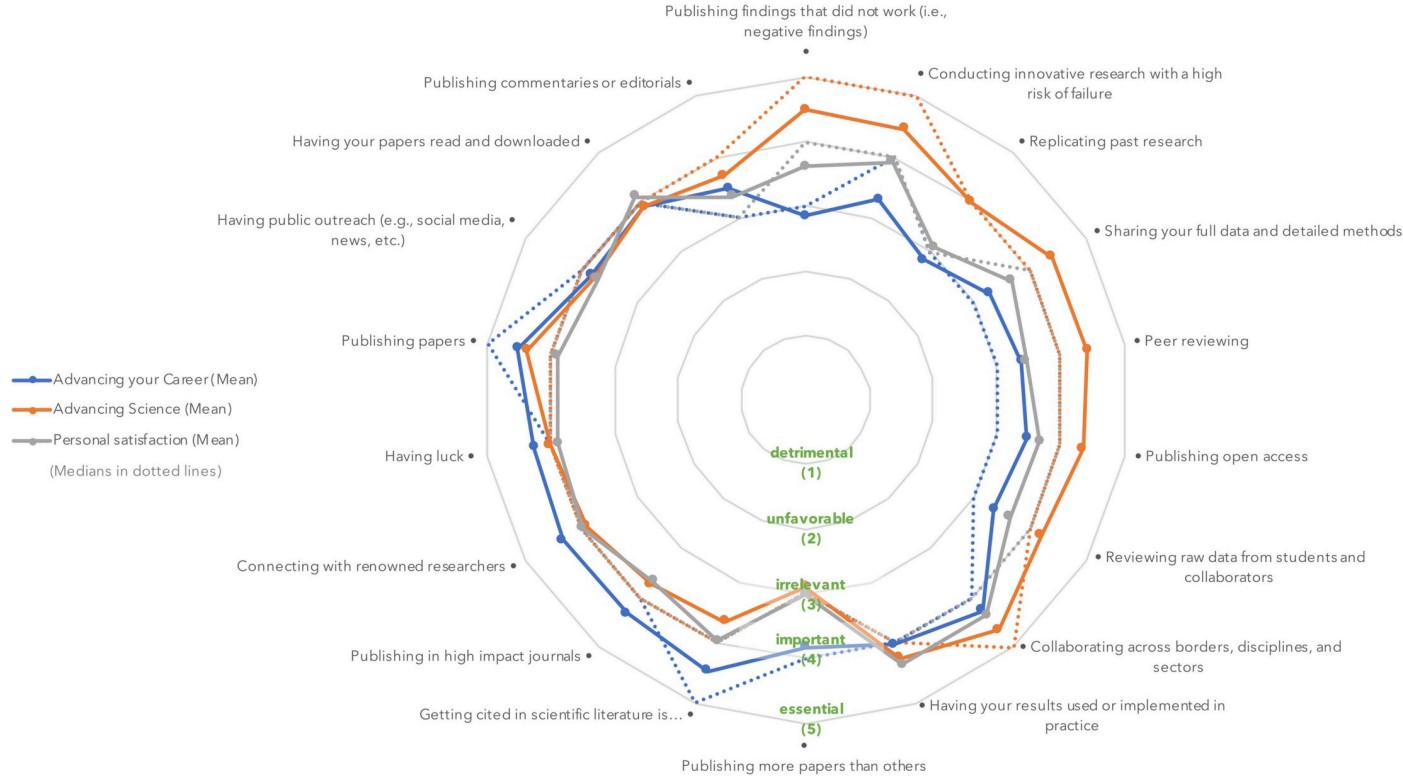

**Fig 2. Mean and median ratings on each dimension for each statement.**

The survey also allowed participants to comment after each item. These comment boxes were rarely used, but the few answers collected provide richer insights about some of the elements included in the survey.

**Publications and metrics.** Comments added to the statements on the importance of 'Publishing papers' proposed that publications were important for science, but were not necessarily used properly. Some comments mentioned that publications are a better indicator of the status and resources of a laboratory than they are of the "*actual research capabilities*" of researchers, while others stated that publishing should aim to share a message, not to increase metrics.

*"It depends on what is in the paper. When we really have something to say, we should say it. On the other hand, publishing for the sake of publishing is very detrimental for science as well as to my personal satisfaction."*

When asked about the impact of 'Publishing in high impact journals', respondents mentioned that expectations of high impact added pressures, but that publishing in high impact journals was also satisfying for researchers since high impact was perceived as a mark of quality.

*"I really believe that it should be irrelevant, but if I am honest, I admit that it is somewhat important to my personal satisfaction. I am proud if I am able to publish a paper (for which I have really done my best) in a good quality journal."*

Others, on the other hand, worried that focusing on high impact journals prevented smaller or local journals from developing.

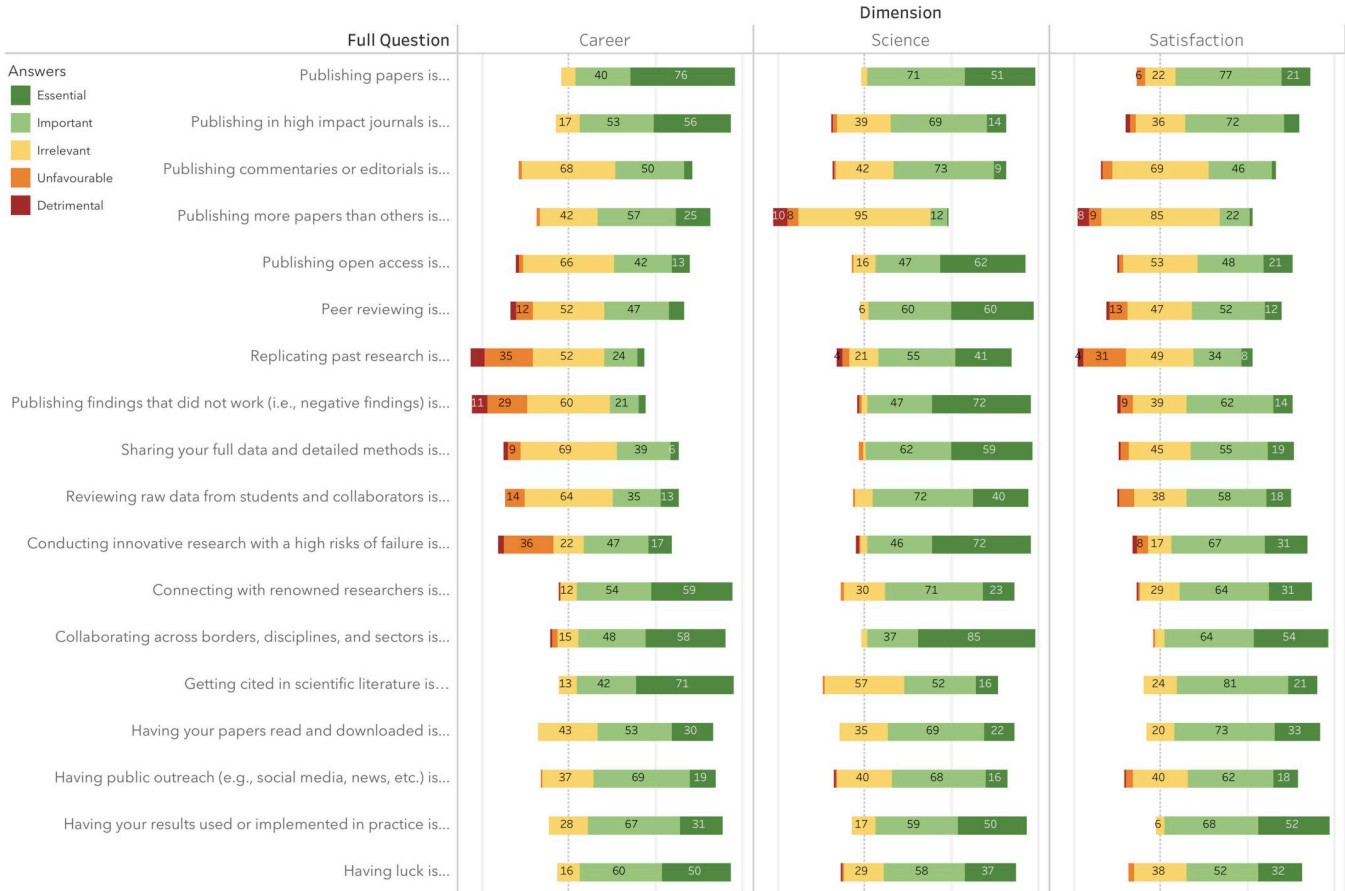

**Fig 3. Distribution of answers for each dimension of each statement.**

**Openness and transparency.** A few statements related to openness also provoked comments. 'Publishing open access' raised a few controversial reactions. One respondent stated being generally "suspicious" of open access journals for asking researchers to cover the publication costs, but most other comments rather mentioned that they would support open access but lacked the funds to do so. Comments on the importance of 'Publishing findings that did not work (i.e., negative results)' explained that negative results must be more visible in published literature—one comment even stated that "*publication bias is the single most detrimental issue to modern science*". Yet respondents recognised that publishing negative results was rarely recognised. One respondent proposed that 'sneaking' negative results into papers with positive results could help counter publication biases. Another respondent, however stated that despite supporting the importance of publishing negative results, her precarious situation—accentuated by gender and seniority inequalities—made it difficult to withhold her convictions and to feel satisfied about publishing negative findings.

"Even though I have been publishing *and* speaking about my failures, I also find it difficult to do so, as I am in a precarious situation as a woman and early career researcher, which of course limits how much I feel personally satisfied . . ."

Finally, the importance of 'Sharing full data and methods' yielded diverging opinions. One respondent disclosed that, having experienced plagiarism in the past, he preferred to share

data in personal discussions than to open it to the world. Others added that sharing raw data implied a lot of extra work and ethical issues they were not ready to deal with. Finally, another three respondents raised doubts about the true benefits of sharing data for promoting integrity.

> "I believe that it is a (naïve) illusion to think that putting full data online is the solution for research misconduct and sloppy research. I can think of (because I often see) many ways to ruin a dataset before it is shared online."

**Reviewing.** From the open comments, we found that some respondents mistrusted the value of peer-review with some believing that it was governed by a principle of "*I scratch your back if you scratch mine*". Yet, given that most respondents rated 'Peer-reviewing' as essential or important in advancing science, doubt on its value may not have been generalized. Indeed, other comments proclaimed that peer review was very important in advancing science in an unselfish way and that researchers "*who publish papers but refuse to invest time in reviewing rig the system*". Beyond the value of peer-review itself, others stated that although they appreciated peer-review, they felt exploited by big publishers when donating their time to it.

> "I like staying up to date and be challenged in my thinking because of reviewing, but I don't like the fact that my free labour helps to increase profit for major publishers."

The same respondent explained that peer review could be a satisfying experience if researchers were "*able to spend time on it during working hours (and not just on top of everything else)*". Along the same lines, responses on the importance of 'Reviewing raw data from students and collaborators' also suggest that reviewing data was important for the quality of the work, but that it would take time which was necessary elsewhere, even if for the wrong reasons.

> "Unfavorable in advancing my career because it takes time and my supervisors want me to spend more time in grant writing. This comment goes for all the activities that really advance science but are difficult to measure. It is the 'doing the best I can' that is not acceptable to my supervisors. They want productivity (in the past this meant getting my name on many publications, nowadays it increasingly means getting grant money)."

## Limitations

Our findings are preliminary and contain important limitations which must be considered when interpreting the results.

The first limitation concerns our recruitment strategy, which was changed and challenged many times to respect the new General Data Protection Regulation. Initially aiming for a controlled recruitment through university contacts and direct emails, we expected to be able to calculate response rates and to have control over the profile of participants who would respond to our survey. Nevertheless, given our inability to reach all research institutions, we shared the survey online to allow for additional institutions to distribute it. Although most broad surveys are now shared using social media and snowballing, this choice inevitably influenced the pool of participants who had access to our survey, and may have increased participation bias. Sharing the survey online also removed the possibility of calculating a clear response rate. Adding to this first concern, our decision to focus largely on the Flemish region of Belgium may have led to answers that are not generalizable elsewhere. Our findings should therefore be interpreted with caution and in consideration of the limited number of principally Flemish participants.

Second, asking participants to estimate their working hours and time distribution relies on precision of recall and accuracy of self-report; two aspects we had no means to verify in the current work. Assessments of the reliability of self-reported working hours are largely absent from the literature. We only found one published paper to support the correlation between recorded and self-reported working hours, but it concerned Japanese workers highly aware of their working schedules [15]. We cannot assume that similar findings would be observed in academic researchers whose working hours vary greatly and whose task concentration changes between academic year periods. Past works and popular surveys investigating the time allocation of scientists found different distribution of work allocation than those we found in our survey, generally reporting a higher proportion of time spent teaching [16, 17]. This difference, which is probably due to the high representation of PhD students among our participants, suggests that our findings might not be representative of other settings and populations and should be interpreted with caution. Despite this limitation, our findings coincide with other works in stating that researchers report working overtime [16, 18–23], that they are subject to heavy administrative burden [22, 24], and that they wish they could dedicate more time to research [25]. Adding to this concern, we also found important to point out that the diversity of profiles included in our study may have impacted the distribution of answers. For example, the amount of time that a PhD student spends writing grants is likely to differ substantially from the amount of time an early career or tenured researcher spends on the same task. In our findings, we aggregated the groups to obtain a general portrait of the time distribution of researchers, but we invite readers to use the data provided if they wish to assess specific differences between seniority profiles.

Another important limitation comes from the formulation of the questions within the survey itself. In our attempt to make a manageable and coherent survey, we preferred to formulate simple questions than to try to describe the full complexity of the terms used. As a result, respondents may have interpreted concepts differently depending on their experience and personal views. We consider that this rich and diverse interpretation of terms and concepts, however, is closer to the reality of research assessments than one in which clear and precise definitions are provided (i.e., evaluation committees rarely have clear definitions of the concepts of 'innovative' or 'excellent' and are generally left to their own interpretations). Nevertheless, we concede that different interpretation of terms may have influenced responses.

Finally, a respondent suggested that "*the categories were not refined enough*" and that we should have included more options or a numerical scale to allow for some nuance. Indeed, our initial idea was to use a slider scale, but given the poor and flimsy rendering of this option on a touchscreen, we opted for pre-defined Likert options. We would probably choose otherwise if we are to pursue this survey further in the future.

## Discussion

In the past few years, research careers raised worrying concerns. Indeed, we know that researchers generally work more hours than they are paid for [22, 23], face high performance pressure [26, 27], are at high risks of stress and mental health issues [23, 28–30], and often experience burnout [31]. This grim portrait of academic careers has raised the alert in the scientific community [e.g., 32–34] and reinforced the need to join efforts in order to address unhealthy research dynamics. One recurrent issue thought to play a key role in this problematic climate is the inadequacy of current research assessments. Indeed, the perceived inadequacy of current research assessments is such that a few organisations and movements have already issued recommendations to encourage changes [e.g., 2–4, 35, 36]. Nevertheless, a hefty debate remains, with some thoroughly approving of the points raised in those recommendations, and others

also finding value in the current methods [5, 37, 38]. In our failure to find and agree on an alternative, research assessments are most often left untouched.

Our findings add to existing insights on the habits, wishes, and perspectives that researchers hold towards research and research assessments. In particular, our results provide a more granular understanding of specific indicators used to assess success in science and detail whether these indicators are believed to help advance science, to help fulfil personal satisfaction, or simply to advance one's career without equally contributing to scientific advancement or personal satisfaction.

## Overworked and still lacking time for research

Almost 80% of full-time researchers who responded to our survey report to work more than 40 hours per week, with 44% stating that they work more than the weekly maximum authorised by the European Union [14]. This finding is no surprise since researchers are known to work overtime and outside office hours [18, 22, 23]. Our findings also reveal that researchers are unsatisfied with the ways in which they need to distribute their time. Respondents wished they could dedicate more of their time to teaching and research, especially to tasks such as 'hands on research work', 'staying up to date', and 'writing papers', a finding that corroborates with similar works [22, 25]. On the other hand, respondents wished they could spend less of their time writing grants and performing other activities such as administration and meetings. This apparent struggle resonates with past works that expose the significant administrative burden of current research careers [16, 17, 24] and with the interviews and focus groups that shaped the current survey in which different research actors argue that the lack of time for research can ultimately lead to a number of issues that jeopardized the integrity and the quality of research [10].

## Lack of reward for openness, transparency, quality, and high risk research

Our findings on specific research assessment indicators provide an overview of the areas which have more importance on advancing science and those which, in turn weigh more on career advancement without necessarily helping to advance science. We were not surprised to find that practices meant to promote openness and transparency (i.e., publishing findings that did not work; sharing your full data and detailed methods; publishing open access), quality (i.e., replicating past research; peer reviewing, and reviewing raw data from students and collaborators), and high risk research (i.e., conducting innovative research with a high risk of failure), were often described to be important or even essential for advancing science, but irrelevant, unfavorable, or sometimes detrimental in advancing one's career. This perspective is shared by several of the important documents and works on research assessments [2–4, 36], and was also thoroughly visible in the qualitative works upon which this survey was built [9, 10]. Following this finding, it seems obvious that research assessments need to stimulate openness and quality, as well as to accept the importance of failure. Nonetheless, our survey also captured nuances which could lead to potential barriers in these areas. First, the distribution of answers and the open comments allowed us to grasp that not everyone is convinced of the value of open access, nor of the added benefit of openly sharing data and methods. Some respondents rated open access as detrimental in advancing science, while some assumed that its profit model based on publication implied bigger biases and lower quality assurance. This finding, which was also echoed in our qualitative interviews [9], highlights that the current views on what constitute best practices are not yet uniform, and that greater and more generalised awareness is needed before customs and cultures can change. Other respondents mentioned that, although they would support open access in theory, the lack of funding for article

processing charges prevented them from publishing in open access journals. Similar issues were noted when discussing publication of negative findings and data sharing, where respondents explained that such tasks come with an added burden and new ethical challenges to which they have no time to dedicate. Consequently, valuing openness in research assessments requires a restructuration that goes far beyond research assessments. If research assessments are to formally value openness, researchers must be given the resources, infrastructures, and potentially even the workforce necessary to undertake such practices without increasing already existing burdens. Valuing openness without providing such resources risks increasing inequalities by further benefiting already successful research groups and disadvantaging young researchers, small institutes, and divergent research fields.

## An overemphasis on competition and prestige

Our findings also help exemplify the overemphasis of current research assessments on competition and prestige. In fact, respondents stated that it was important for their career to publish more papers than others, to publish in high impact journals, to be cited, and to have a strong network of renowned researchers. These indicators, however, were said to be of lesser importance in advancing science and in contributing to respondents' personal satisfaction. In today's academia, researchers are expected to be excellent, yet their excellence is only recognized if they are highly productive, visible, and impactful, three characteristics which, when added to the scarcity of senior positions available [39, 40], nurture very competitive climates [41]. The tight competition forces researchers to spend a lot of their research time writing grants to compensate for chances of success which are often negligible [42]. In turn, the colossal demands for research money also adds pressure to the funders who face an overload of applications to revise [10]. Paradoxically however, funders also need to ask researchers to peer-review and judge applications, further reducing the time that researchers have available for conducting good research.

## Conclusion

Our survey grasps the perspective of researchers on the value that different research activities have in advancing science, in advancing research careers, and in contributing to researchers' personal satisfaction. We found that respondents would like to be able to dedicate more time on direct research activities such as writing papers and performing hands-on research work, and wish they could dedicate less time to writing grants and other tasks such as meetings and administration. Our survey also reveals that many research practices related to openness, transparency, quality, and acceptance of failure are perceived as important or even essential in advancing science, but are seen as irrelevant or even sometimes detrimental in advancing researchers' careers. Conversely, some practices which inflate the prestige, visibility, and competitiveness of researchers are seen as important in career assessments, but much less relevant in advancing science. It is important to consider that our survey captured the perspectives of a limited sample of predominantly Flemish researchers and may thus be of limited generalisability. Nonetheless, our findings align with a growing body of international works, declarations, and reports on the topic [see for example 2, 3, 35, 36, 43, 44]. Together with this growing body of literature, our findings support that research assessments need to be addressed so that researchers' careers consider activities that pursue the genuine advancement of science. Yet, our findings also show that there are nuances and disagreements on the impact of specific practices in advancing science. To ensure that changes to research assessments benefit, rather than worsen, research practices and researchers' working conditions, a thorough restructuration of the resources and infrastructures needs to take place. Beyond recognizing the importance of

openness, transparency, and quality, institutions and funders should work together to enable the establishment of local resources that assist and support researchers in fostering these values.

## Supporting information

**S1 File.**
(PDF)

**S2 File.**
(PDF)

**S3 File.**
(CSV)

**S1 Table.**
(PDF)

## Acknowledgments

Contributors

The authors would like to thank several contributors to the present project *Methodology*: Vincent Larivière provided guidance on the survey content, terminology, and recruitment methods; Patricia Tielens helped us understand how to distribute the survey in respect with GDPR; Raymond De Vries, Søren Holm, and Daniele Fanelli provided feedback on earlier versions of the survey, Dana Hawwash, Paolo Corsico, and Audrey Wolff provided feedback on final versions of the survey. *Resources*: Many people helped share the survey, thereby helping us find participants. These include Deans, Directors of doctoral school, and secretaries of the universities we contacted, Raffaella Ravinetto, Hannelore Storms, Carl Lachat, Stefanie Van der Burght, and many more people who shared, re-tweeted, or talked about our survey to colleagues. We also wish to thank the participants themselves for their time, efforts, and for their willingness to share their thoughts. *Formal analysis*: Geert Molenberghs provided guidance and continued support on the appropriate statistical analyses to use. *Peer-review*: The authors would also like to thank Ludo Waltman for his excellent review which helped us correct important aspects of our initial manuscript.

## Author Contributions

**Conceptualization:** Noémie Aubert Bonn, Wim Pinxten.

**Data curation:** Noémie Aubert Bonn.

**Formal analysis:** Noémie Aubert Bonn.

**Funding acquisition:** Wim Pinxten.

**Investigation:** Noémie Aubert Bonn.

**Methodology:** Noémie Aubert Bonn, Wim Pinxten.

**Project administration:** Noémie Aubert Bonn, Wim Pinxten.

**Resources:** Noémie Aubert Bonn, Wim Pinxten.

**Supervision:** Wim Pinxten.

**Validation:** Noémie Aubert Bonn, Wim Pinxten.

**Visualization:** Noémie Aubert Bonn.

**Writing – original draft:** Noémie Aubert Bonn.

**Writing – review & editing:** Noémie Aubert Bonn, Wim Pinxten.

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
