## [Decision Letter · Decision Letter 0]

21 Aug 2020

PONE-D-20-19974

Advancing science or advancing careers? Researchers' opinions on success indicators

PLOS ONE

Dear Dr. Aubert Bonn,

Thank you for submitting your manuscript to PLOS ONE. After careful consideration, we feel that it has merit but does not fully meet PLOS ONE’s publication criteria as it currently stands. Therefore, we invite you to submit a revised version of the manuscript that addresses the points raised during the review process.

Please change the discussion of the results and implications according to the notes of the reviewer (see paragraph starting with "In the final section, the authors draw a few quite strong ..."

Also, present all numerical estimates for parameters with just the SIGNIFICANT digits. Which digits are significant should be apparent from CI for estimates.

We look forward to receiving your revised manuscript.

Kind regards,

Luís A. Nunes Amaral, Ph.D.

Academic Editor

PLOS ONE

Journal Requirements:

2. Please include additional information regarding the survey or questionnaire used in the study and ensure that you have provided sufficient details that others could replicate the analyses.

For instance, if you developed a questionnaire as part of this study and it is not under a copyright license more restrictive than CC-BY, please include a copy, in both the original language and English, as Supporting Information.

4. Your ethics statement must appear in the Methods section of your manuscript. If your ethics statement is written in any section besides the Methods, please move it to the Methods section and delete it from any other section. Please also ensure that your ethics statement is included in your manuscript, as the ethics section of your online submission will not be published alongside your manuscript.

Reviewers' comments:

Reviewer's Responses to Questions

**Comments to the Author**

1. Is the manuscript technically sound, and do the data support the conclusions?

Reviewer #1: Yes

2. Has the statistical analysis been performed appropriately and rigorously? 

Reviewer #1: Yes

3. Have the authors made all data underlying the findings in their manuscript fully available?

Reviewer #1: Yes

4. Is the manuscript presented in an intelligible fashion and written in standard English?

Reviewer #1: Yes

5. Review Comments to the Author

Reviewer #1: I recommend to accept this paper for publication after the authors have made some small improvements.

On p. 8, the authors state that “125 participants completed the survey”. However, according to Table 2, there are 126 participants.

“On average, respondents who declared working full time worked 46.91 hours per week (median 46) but the distribution was very wide (Table 3)” (p. 8): Table 3 doesn’t show that the distribution is very wide. The table shows a maximum value, but this is insufficient to conclude that a distribution is very wide.

“we found that participants wished they could dedicate more time to ‘Teaching’ (paired t(92) = 3.3539, p = 0.001159) and especially to ‘Research’ (paired t(92) = 4.2818, p = 4.545e-05)” (p. 10): I don’t understand the result for teaching. Looking at the results for teaching in Figure 1, it seems that ‘reality’ and ‘ideal’ more or less coincide, and if there is a difference, it seems that ‘ideal’ is below ‘reality’, not above.

“Almost 80% of full-time researchers who responded to our survey worked more than 40 hours per week” (p. 15): This statement is not entirely accurate. It would be more accurate to write: “Almost 80% of full-time researchers who responded to our survey report to work more than 40 hours per week”. I wonder whether people are able to accurately estimate the number of hours they work per week. Personally I find it quite difficult to estimate this. It could be that people systematically overestimate or underestimate the number of hours they work. Perhaps the authors could check whether there is any literature in which the accuracy of these types of survey responses is investigated.

In the final section, the authors draw a few quite strong conclusions based on their survey (e.g., “a thorough restructuration of the resources and infrastructures needs to take place”). While these conclusions seem reasonable to me, I think the authors need to acknowledge that the survey provides only limited evidence to support these conclusions. There are various reasons why the evidence provided by the survey is limited. First, the number of survey participants is quite limited. Second, there probably is a participation bias. Third, most respondents are from Flanders, so the survey can be used to draw conclusions only about Flanders. My recommendation to the authors is to emphasize how the survey results align with the broader literature. In this way, the authors can make clear that their conclusions are supported not only by their survey results, but also by the broader literature.

The authors sometimes present numerical results with lots of decimals (e.g., bottom paragraph on p. 10). There is no need to report so many decimals. These decimals are not informative. My recommendation is to reduce the number of decimals that are reported.

Following initiatives to improve statistical reporting (e.g., https://en.wikipedia.org/wiki/Estimation_statistics), I believe it would be preferable to focus more strongly on measures of effect size (e.g., the mean difference in the answers given to different survey questions), complemented with confidence intervals, instead of using null hypothesis significance testing (NHST). NHST has the disadvantage of promoting dichotomous ways of thinking. The magnitude of an effect often doesn’t get proper attention in NHST. Since there is no agreement on the pros and cons of different statistical methods, I consider the use of NHST to be acceptable, but I hope the authors will consider using more appropriate statistical methods in the future.

6. PLOS authors have the option to publish the peer review history of their article (what does this mean?). If published, this will include your full peer review and any attached files.

Reviewer #1: **Yes: **Ludo Waltman

---

## [Author Response · Author response to Decision Letter 0]

4 Oct 2020

**Note: I highly recommend that reviewers and editors use the PDF version of this response since the formatting helps understand where the reviewer's comments end and where our responses begin.**

INTRODUCTORY NOTE FROM THE AUTHORS:

Dear Dr. Nunes Amaral, dear Prof. Waltman,

Thank you very much for your careful review of our manuscript. I really appreciate the excellent comments of Prof. Waltman, and I thoroughly adapted the manuscript to fit all points raised.

Being a PhD student with little experience in reporting confidence intervals however, I wanted to make sure that I was doing the changes correctly. For this reason, I detail what I did below, and would be grateful if I could receive a bit of feedback on my adaptations to feel comfortable that our findings are reported appropriately.

In short, I re-ran the analyses to capture the confidence intervals and added those throughout to a new results table (Table 4) and to the Supplementary table S4. For the time analyses I grabbed the lower bound and the upper bound directly from R and rounded them to second digit. For the dimensions analyses, I took the confidence intervals that are reported alongside the ‘Pairwise Comparisons’ (i.e., the post hoc tests) in SPSS, and reported them in the Supplementary file S4 keeping all three digits reported (I could have reduced to 2 digits but I was unsure what was best). Throughout the manuscript, I removed statistical data and referred readers to Table 4 or Supplementary Table S4. I rounded p values to <0.05, <0.01, and <0.001 to avoid the false sense of precision, and I added the corresponding significance (*, **, ***, respectively) in the new Fig. 1. The reason I took out all values from the text was because I was concurrently adapting the manuscript to reviewers of my thesis who suggested the change. I am happy to adapt to whatever you find best. 

Please let me know if there is anything I could have done better. I am happy to edit the manuscript further (that is, after my thesis defence of the 14th!).

I sincerely thank both of you for your time and efforts. I am learning in the process of this project, and I feel very fortunate to benefit from your expertise. I also added a note to thank Prof. Waltman in the acknowledgements, if he prefers not to be mentioned I can take it out.

I look forward to hearing back from you.

Kind regards,

Noémie Aubert Bonn (Corresponding author)

 

PONE-D-20-19974

Advancing science or advancing careers? Researchers' opinions on success indicators

PLOS ONE

Dear Dr. Aubert Bonn,

Thank you for submitting your manuscript to PLOS ONE. After careful consideration, we feel that it has merit but does not fully meet PLOS ONE’s publication criteria as it currently stands. Therefore, we invite you to submit a revised version of the manuscript that addresses the points raised during the review process.

Please change the discussion of the results and implications according to the notes of the reviewer (see paragraph starting with "In the final section, the authors draw a few quite strong ..."

RESPONSE FROM THE AUTHORS: Thank you, I have changed the text according to the reviewer’s excellent comments. My specific changes are detailed below and visible in the manuscript that contains track changes.

Also, present all numerical estimates for parameters with just the SIGNIFICANT digits. Which digits are significant should be apparent from CI for estimates.

RESPONSE FROM THE AUTHORS: I have adapted the reporting of statistical values, but am unsure if my reporting of significant digits (as mentioned above). I hope that I did it correctly, and am very happy to edit as needed.

We look forward to receiving your revised manuscript.

Kind regards,

Luís A. Nunes Amaral, Ph.D.

Academic Editor

PLOS ONE

Journal Requirements:

2. Please include additional information regarding the survey or questionnaire used in the study and ensure that you have provided sufficient details that others could replicate the analyses.

For instance, if you developed a questionnaire as part of this study and it is not under a copyright license more restrictive than CC-BY, please include a copy, in both the original language and English, as Supporting Information.

4. Your ethics statement must appear in the Methods section of your manuscript. If your ethics statement is written in any section besides the Methods, please move it to the Methods section and delete it from any other section. Please also ensure that your ethics statement is included in your manuscript, as the ethics section of your online submission will not be published alongside your manuscript.

Reviewers' comments:

Reviewer's Responses to Questions

Comments to the Author

1. Is the manuscript technically sound, and do the data support the conclusions?

Reviewer #1: Yes

2. Has the statistical analysis been performed appropriately and rigorously? 

Reviewer #1: Yes

3. Have the authors made all data underlying the findings in their manuscript fully available?

Reviewer #1: Yes

4. Is the manuscript presented in an intelligible fashion and written in standard English?

Reviewer #1: Yes

5. Review Comments to the Author

Reviewer #1: I recommend to accept this paper for publication after the authors have made some small improvements.

On p. 8, the authors state that “125 participants completed the survey”. However, according to Table 2, there are 126 participants.

RESPONSE FROM THE AUTHORS: Thank you for noticing this mistake, it should indeed read 126.

“On average, respondents who declared working full time worked 46.91 hours per week (median 46) but the distribution was very wide (Table 3)” (p. 8): Table 3 doesn’t show that the distribution is very wide. The table shows a maximum value, but this is insufficient to conclude that a distribution is very wide.

RESPONSE FROM THE AUTHORS: Correct. I added the minimum values to Table 3 so the range can be observed.

“we found that participants wished they could dedicate more time to ‘Teaching’ (paired t(92) = 3.3539, p = 0.001159) and especially to ‘Research’ (paired t(92) = 4.2818, p = 4.545e-05)” (p. 10): I don’t understand the result for teaching. Looking at the results for teaching in Figure 1, it seems that ‘reality’ and ‘ideal’ more or less coincide, and if there is a difference, it seems that ‘ideal’ is below ‘reality’, not above.

RESPONSE FROM THE AUTHORS: I am very grateful that you spotted this and pointed it out to me. It took me some time to understand what was happening, but I then realized that the format of Fig. 1 as I submitted it was incorrect (it only considered unique values and therefore reported incorrect medians and standard deviations). I have now rectified this problem and have thoroughly updated Fig. 1. I also added a green line to represent the mean for each category to allow better comparison. I double-checked the accuracy of the reported means and medians. I hope the new figure is clearer and more informative, and I sincerely thank you again for spotting this issue before publication.

“Almost 80% of full-time researchers who responded to our survey worked more than 40 hours per week” (p. 15): This statement is not entirely accurate. It would be more accurate to write: “Almost 80% of full-time researchers who responded to our survey report to work more than 40 hours per week”. I wonder whether people are able to accurately estimate the number of hours they work per week. Personally I find it quite difficult to estimate this. It could be that people systematically overestimate or underestimate the number of hours they work. Perhaps the authors could check whether there is any literature in which the accuracy of these types of survey responses is investigated.

RESPONSE FROM THE AUTHORS: Absolutely. I adapted the text to capture this distinction, and also changed the title of Fig. 1 to “Self-reported percentage of time”. I also looked for some literature on the topic and added the following paragraph to the Limitations section:

“Second, asking participants to estimate their working hours and time distribution relies on precision of recall and accuracy of self-report; two aspects we had no means to verify in the current work. Assessments of the reliability of self-reported working hours are largely absent from the literature. We only found one published paper to support the correlation between recorded and self-reported working hours, but it concerned Japanese workers highly aware of their working schedules (14). We cannot assume that similar findings would be observed in academic researchers whose working hours vary greatly and whose task concentration changes between academic year periods. Past works and popular surveys investigating the time allocation of scientists found different distribution of work allocation than those we found in our survey, generally reporting a higher proportion of time spent teaching (15, 16). This difference, which is probably due to the high representation of PhD students among our participants, suggests that our findings might not be representative of other settings and populations and should be interpreted with caution. Despite this limitation, our findings coincide with other works in stating that researchers report working overtime (15, 17-21), that they are subject to heavy administrative burden (21, 22), and that they wish they could dedicate more time to research (23).

14. Imai T, Kuwahara K, Miyamoto T, Okazaki H, Nishihara A, Kabe I, et al. Validity and reproducibility of self-reported working hours among Japanese male employees. J Occup Health. 2016;58(4):340-6.

15. Ziker J. The Blue Review. 2014 31 March. Available from: https://www.boisestate.edu/bluereview/faculty-time-allocation/.

16. Matthews D. If you love research, academia may not be for you. Times Higher Education. 2018 8 November. Available from: https://www.timeshighereducation.com/blog/if-you-love-research-academia-may-not-be-you.

17. Barnett A, Mewburn I, Schroter S. Working 9 to 5, not the way to make an academic living: observational analysis of manuscript and peer review submissions over time. BMJ. 2019;367.

18. Powell K. Young, talented and fed-up: scientists tell their stories. Nature. 2016;538:446-9.

19. Bothwell E. Work-life balance survey 2018: long hours take their toll on academics. Times Higher Education. 2018 8 February. Available from: https://www.timeshighereducation.com/features/work-life-balance-survey-2018-long-hours-take-their-toll-academics.

20. Mckenna L. How Hard Do Professors Actually Work? The Atlantic. 2018 7 February. Available from: https://www.theatlantic.com/education/archive/2018/02/how-hard-do-professors-actually-work/552698/.

21. Koens L, R., Jonge HaJd. What motivates researchers? Research excellence is still a priority. The Hague: Rathenau Instituut; 2018. Available from: https://www.rathenau.nl/sites/default/files/2018-07/What%20motivates%20researchers.pdf.

22. Schneider SL, Ness KK, Shaver K, Brutkiewicz R. Federal Demonstration Partnership 2012 Faculty Workload Survey - Research Report. 2014. Available from: https://osp.od.nih.gov/wp-content/uploads/SMRB_May_2014_2012_Faculty_Workload_Survey_Research_Report.pdf.

23. Mergaert L, Raeymaekers P. Researchers at Belgian Universities: What drives them? Which obstacles do they encounter? : Kind Baudouin Foundation; 2017. Report No.: ISBN: D/2893/2017/16. Available from: https://www.kbs-frb.be/en/Virtual-Library/2017/20171113PP.”

In the final section, the authors draw a few quite strong conclusions based on their survey (e.g., “a thorough restructuration of the resources and infrastructures needs to take place”). While these conclusions seem reasonable to me, I think the authors need to acknowledge that the survey provides only limited evidence to support these conclusions. There are various reasons why the evidence provided by the survey is limited. First, the number of survey participants is quite limited. Second, there probably is a participation bias. Third, most respondents are from Flanders, so the survey can be used to draw conclusions only about Flanders. My recommendation to the authors is to emphasize how the survey results align with the broader literature. In this way, the authors can make clear that their conclusions are supported not only by their survey results, but also by the broader literature.

RESPONSE FROM THE AUTHORS: Thank you for noticing this point. I have added additional examples from the literature within the discussion, and I have toned down the conclusion by noting the following: 

“It is important to consider that our survey captured the perspectives of a limited sample of predominantly Flemish researchers and may thus be of limited generalisability. Nonetheless, our findings align with a growing body of international works, declarations, and reports on the topic (see for example 2, 3, 35, 36, 43, 44). Together with this growing body of literature, our findings support that research assessments need to be addressed so that researchers' careers consider activities that pursue the genuine advancement of science.”

I am happy to adapt it further if deemed necessary.

The authors sometimes present numerical results with lots of decimals (e.g., bottom paragraph on p. 10). There is no need to report so many decimals. These decimals are not informative. My recommendation is to reduce the number of decimals that are reported.

RESPONSE FROM THE AUTHORS: As mentioned in the introductory text, I have moved all statistical results from this section to Table 4. I only keot two decimals and have simplified the p values throughout to p < 0.05, p < 0.01, and p < 0.001. Note that I left all three decimals in Supplementary Table S4. I would be happy to implement any changes that seems fit.

Following initiatives to improve statistical reporting (e.g., https://en.wikipedia.org/wiki/Estimation_statistics), I believe it would be preferable to focus more strongly on measures of effect size (e.g., the mean difference in the answers given to different survey questions), complemented with confidence intervals, instead of using null hypothesis significance testing (NHST). NHST has the disadvantage of promoting dichotomous ways of thinking. The magnitude of an effect often doesn’t get proper attention in NHST. Since there is no agreement on the pros and cons of different statistical methods, I consider the use of NHST to be acceptable, but I hope the authors will consider using more appropriate statistical methods in the future.

RESPONSE FROM THE AUTHORS: As mentioned in the ‘Introductory note from the authors’ above, we are learning in this process and are grateful to learn the best methods. I re-ran the analyses to capture the confidence intervals and added those throughout Table 4 and the Supplementary table S4. As noted above, one of the jury of my thesis suggested that it may be better to take out the statistical analyses entirely from this manuscript, and this explains why I moved the results to Table 4. I would appreciate your views on this matter. 

6. PLOS authors have the option to publish the peer review history of their article (what does this mean?). If published, this will include your full peer review and any attached files.

Do you want your identity to be public for this peer review? For information about this choice, including consent withdrawal, please see our Privacy Policy.

Reviewer #1: Yes: Ludo Waltman

---

## [Decision Letter · Decision Letter 1]

12 Nov 2020

PONE-D-20-19974R1

Advancing science or advancing careers? Researchers' opinions on success indicators

PLOS ONE

Dear Dr. Aubert Bonn,

Thank you for submitting your manuscript to PLOS ONE. After careful consideration, we feel that it has merit but does not fully meet PLOS ONE’s publication criteria as it currently stands. Therefore, we invite you to submit a revised version of the manuscript that addresses the points raised during the review process.

As we discussed over email, please revise the reporting of the results in Table 4 according to the procedure we agreed upon.

We look forward to receiving your revised manuscript.

Kind regards,

Luís A. Nunes Amaral, Ph.D.

Academic Editor

PLOS ONE

Reviewers' comments:

Reviewer's Responses to Questions

**Comments to the Author**

1. If the authors have adequately addressed your comments raised in a previous round of review and you feel that this manuscript is now acceptable for publication, you may indicate that here to bypass the “Comments to the Author” section, enter your conflict of interest statement in the “Confidential to Editor” section, and submit your "Accept" recommendation.

Reviewer #1: (No Response)

2. Is the manuscript technically sound, and do the data support the conclusions?

Reviewer #1: Yes

3. Has the statistical analysis been performed appropriately and rigorously? 

Reviewer #1: No

4. Have the authors made all data underlying the findings in their manuscript fully available?

Reviewer #1: Yes

5. Is the manuscript presented in an intelligible fashion and written in standard English?

Reviewer #1: Yes

6. Review Comments to the Author

Reviewer #1: Thank you for the careful revision of your paper. I consider your work to be acceptable for publication. However, I find it hard to understand the confidence intervals reported in Table 4. I recommend to remove them from your paper.

7. PLOS authors have the option to publish the peer review history of their article (what does this mean?). If published, this will include your full peer review and any attached files.

Reviewer #1: **Yes: **Ludo Waltman

---

## [Author Response · Author response to Decision Letter 1]

23 Nov 2020

Response to editor and reviewer

15 November 2020

Manuscript PONE-D-20-19974R1

Dear Luís A. Nunes Amaral, Dear Prof. Waltman,

I wish to thank you both for your help and patience, and for allowing me learn how to better report my findings. I have now adapted the manuscript and accompanying files to reflect the changes we discussed. My changes are as follows:

Changes to the manuscript: 

- I corrected Table 4 by performing all analyses without the log correction. Most results remained the same, but the difference between real and ideal times spent ‘Writing grants’ became significant as a result of the change (i.e., respondents wish they could spend less time writing grants). I believe that this is due to the skewed response curve in the time spent writing grant, possibly because of the different seniority profiles included in the study. For instance, PhD students are likely to spend very little time writing grants, while early career researchers sometime spend the majority of their time doing it. While the log transformation would control for this skewed distribution of answers, the new findings do not and the means are now interpreted as significantly different. To reflect this finding, I added a small section in the Limitation section stating 

“…we also found important to point out that the diversity of profiles included in our study may have impacted the distribution of answers. For example, the amount of time that a PhD student spends writing grants is likely to differ substantially from the amount of time an early career or tenured researcher spends on the same task. In our findings, we aggregated the groups to obtain a general portrait of the time distribution of researchers, but we invite readers to use the data provided if they wish to assess specific differences between seniority profiles.

- I adapted the rest of the text to reflect the new finding about the time spent writing grants

- I also realised that I did not mention the statistical softwares used and added those to the Methods-Analysis section. 

- Finally, I did a few changes to quotation marks (‘ vs “) since I noticed that I was not consistent in their use.

Changes to figures and accompanying files:

- I adapted Figure 1 to reflect new statistical significance (*, **, ***)

- I adapted Supplementary file S2 in which I report the Code used to reflect the new statistical code used. In doing so, I also added a small section to create the files to be used in the analyses from the supplementary data file so that the results are easily reusable.

I look forward to hearing back from you.

Kind regards,

Noémie

---

## [Editor Report · Decision Letter 2]

25 Nov 2020

Advancing science or advancing careers? Researchers' opinions on success indicators

PONE-D-20-19974R2

Dear Dr. Aubert Bonn,

We’re pleased to inform you that your manuscript has been judged scientifically suitable for publication and will be formally accepted for publication once it meets all outstanding technical requirements.

Kind regards,

Luís A. Nunes Amaral, Ph.D.

Academic Editor

PLOS ONE
---

## [Editor Report · Acceptance letter]

29 Jan 2021

PONE-D-20-19974R2 

Advancing science or advancing careers? Researchers' opinions on success indicators 

Dear Dr. Aubert Bonn:

I'm pleased to inform you that your manuscript has been deemed suitable for publication in PLOS ONE. Congratulations! Your manuscript is now with our production department. 

Kind regards, 

on behalf of

Dr. Luís A. Nunes Amaral 

Academic Editor

PLOS ONE